# Assessment of coronary spasms with transluminal attenuation gradient in coronary computed tomography angiography

Jae Yang Park[1], Eun-Ju Kang[1]*, Moo Hyun Kim[2], Hwan Seok Yong[3], Seung-Woon Rha[4]

1 Department of Radiology, College of Medicine, Dong-A University, Busan, Republic of Korea,
2 Department of Cardiology, College of Medicine, Dong-A University, Busan, Republic of Korea,
3 Department of Radiology, Korea University Guro Hospital, Seoul, Republic of Korea, 4 Department of Cardiology, Korea University Guro Hospital, Seoul, Republic of Korea

* medcarrot@dau.ac.kr

## Abstract

### Purpose

To evaluate the imaging features of coronary spasm, including transluminal attenuation gradient (TAG) on coronary computed tomography angiography (CCTA), in patients with vasospastic angina (VA).

### Methods

A total of 43 patients with a high clinical likelihood of VA were included in the study. All the subjects underwent double CCTA acquisition: CCTA without a vasodilator ('baseline CT') and CCTA during continuous intravenous nitrate infusion ('IV nitrate CT'). A catheterized ergonovine provocation test was used to determine true VA patients. Coronary spasm is classified into focal- and diffuse-types according to morphological differences. We measured TAG and contrast enhancement of the proximal ostium (ProxHU) of each coronary artery for both the baseline and IV nitrate CT.

### Results

Twenty-four patients (55.8%) showed positive results of coronary vasospasm on the provocation test. Thirty-eight vessels showed coronary spasms (29.5%): Focal-type in nine vessels (24%), and diffuse-type in 29 (76%). In the baseline CT, LCX showed significantly lower (steeper) TAG in spasm(+) vessels than in spasm(-) vessels, while LAD and RCA showed no significant differences in TAG. The ProxHU of LAD showed significantly lower values in spasm(+) vessels than in spasm(-) vessels, while the other vessels did not show significant differences in ProxHU. For IV nitrate CT, there were no significant differences in either the TAG and ProxHU between spasm(+) and (-) vessels for all the three vessel types. In subgroup analysis for spasm(+) vessels, diffuse spasms showed significantly lower TAG than focal spasms, while the ProxHU did not differ between the two types of spasm.

**Data Availability Statement:** All relevant data are within the paper, figures, and its Supporting Information files.

**Funding:** This study was supported by research funds from Dong-A University.

**Competing interests:** The authors have declared that no competing interests exist.

## Conclusions

A relatively large percentage of coronary spasms present as diffuse type, and the TAG values significantly differed according to the morphological type of the coronary spasm.

## Introduction

Coronary artery spasm is a frequent cause of acute chest pain. It can cause angina pectoris, various ischemic diseases such as acute myocardial infarction, and even sudden cardiac death [1–4]. Catheterized coronary angiography (CAG) with a provocation test using acetylcholine or ergonovine is essential for diagnosing coronary artery spasms [5]. However, this procedure is invasive and involves a potential risk of severe myocardial ischemia or arrhythmia. Therefore, there is a need for less invasive diagnostic methods for coronary spasms. Coronary computed tomography (CT) angiography (CCTA) using multidetector CT has been widely used as a noninvasive imaging technique for evaluating coronary artery disease [6,7]. However, coronary spasms transiently occur during rest, especially in the early morning, and rarely during the day. Further, sublingual vasodilators are routinely administered before CCTA for coronary artery dilation. The sensitivity of CCTA is considerably lower as compared to its specificity. According to a study by Kang et al., the sensitivity of CCTA was 48% [8]. Taken together, this suggests that CCTA is inadequate as an initial screening tool for VA diagnosis. Previously, we adapted a double-acquisition CCTA protocol, which acquires CT at two different time points in the same patient. The first one is the baseline CT, performed in the early morning without a vasodilator, and the second is performed a few hours later using a vasodilator: a continuous intravenous nitrate infusion CT (IV nitrate CT) [9]. This protocol yielded a relatively high sensitivity (73%) for VA diagnosis [8,10]. However, this sensitivity was insufficient for predicting coronary spasms. Moreover, the CT protocol requires CCTA to be performed twice for each individual, which resulted in the use of more contrast medium and a higher radiation dose than that in single-acquisition protocols.

Transluminal attenuation gradient (TAG), which is a linear regression coefficient between the axial distance and luminal attenuation, allows the functional analysis of coronary stenosis without additional radiation exposure or use of contrast medium [11–13]. However, clinical validation studies have reported conflicting results of the usefulness of using TAG at determining coronary arterial flow [14,15], since TAG may be affected by changes in coronary luminal diameter and collateral vascular enhancement.

A focal type coronary spasm may occur as a focal stenotic lesion with negative remodeling; however, the distal vessel diameter is often normal. In the case of a diffuse type coronary spasm, the vessel shows a diffuse small diameter throughout the whole single vessel without normal diameter tapering. Therefore, diameter changes might not affect TAG values in both types of spasm. Moreover, collateral vessels, which develop in organic atherosclerotic lesions, do not develop in coronary spasms due to the transient occurrence and resolution of coronary spasms. To our knowledge, there have been no studies on TAG in coronary spasms. We aimed to examine the imaging features of coronary spasms, including TAG in CCTA, in patients with vasospastic angina (VA). Moreover, we aimed to evaluate differences in TAG between focal and diffuse coronary spasms.

## Materials and methods

### Subjects

We retrospectively reviewed 57 consecutive patients aged 30–73 years with a high clinical likelihood of VA between March 2017 and April 2019. All the subjects were part of the Dual-

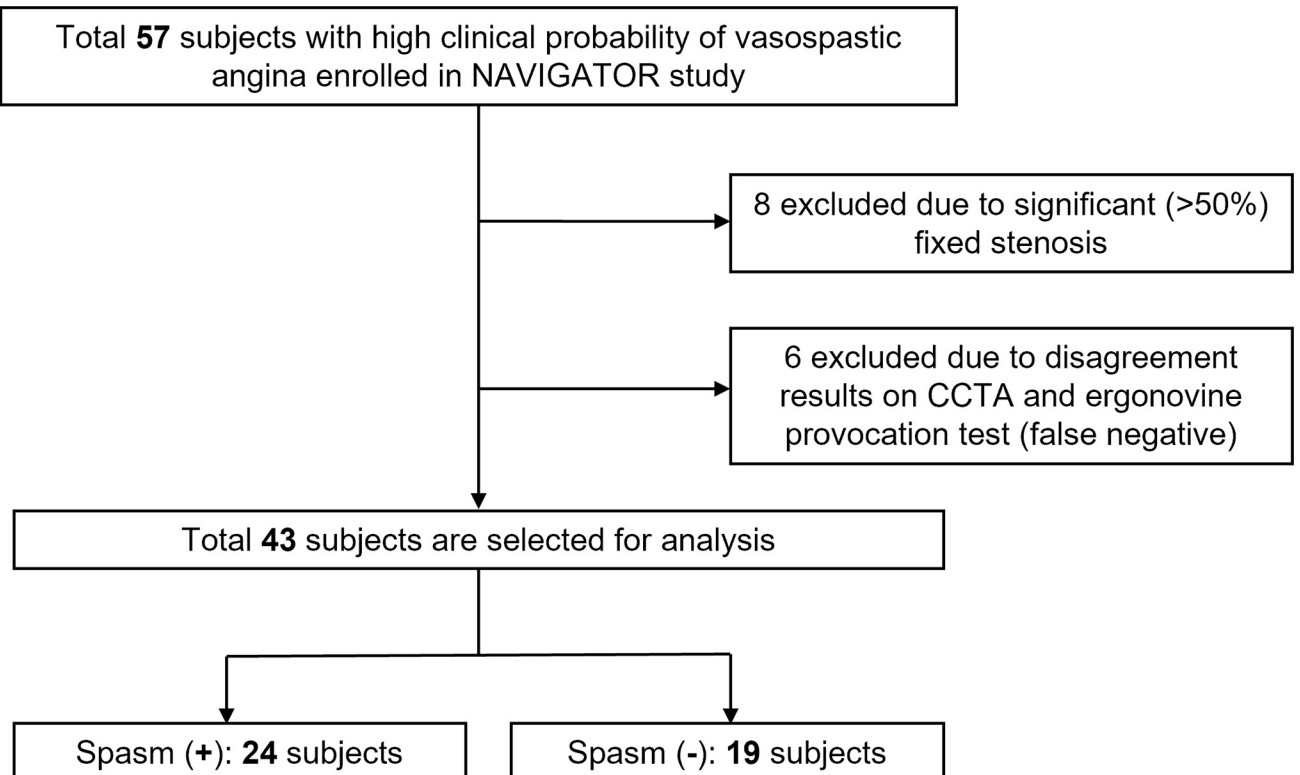

**Fig 1. Patient flow diagram.** We excluded 14 patients from the initial 57 patients due to significant fixed stenosis (n = 8) or inconsistency between the CCTA and spasm provocation test (n = 6). Finally, 43 patients were selected for analysis. CCTA, coronary computed tomography angiography.

acquisition of Noninvasive Cardiac Imaging in Vasospastic Angina Korean Registry (NAVI-GATOR study) [9,16]. These patients underwent baseline CCTA without a vasodilator ('baseline CT') early morning, followed by catheterized CAG with an ergonovine provocation test. Subsequently, they underwent repetitive CCTA during a continuous intravenous (IV) nitrate infusion ('IV nitrate CT') within 3 days.

Since TAG could be influenced by the vessel diameter and length, we included only right-dominant patients i.e., patients whose posterior descending artery is supplied by the right coronary artery (RCA)) [17]. We excluded eight patients with significant fixed stenosis, that was defined as ≥50% stenosis compared with the adjacent non-diseased arterial segment on conventional CAG. The diagnostic performance of CCTA for the detection of coronary spasm showed that the sensitivity, specificity, positive predictive value, negative predictive value, and accuracy were 80%, 100%, 100%, 76%, and 87.76%, respectively. Per-vessel analysis results are shown in S1 Table.

For the analysis of TAG values on CCTA, we excluded 6 patients who showed inconsistent results on CCTA (negative) and the ergonovine provocation test (positive). Finally, a total of 43 patients (spasm(+) patients = 24, spasm(-) patients = 19) were enrolled in this study (Fig 1).

This retrospective multi-center study was approved by our institutional ethics committee, which waived the requirement for formal informed consent.

## Catheterized ergonovine provocation test

VA was diagnosed based on findings from an invasive CAG and a positive result in the ergonovine provocation test [18]. Experienced cardiologists performed invasive CAG via radial

access using a single coronary artery diagnostic catheter on the left coronary artery initially, followed by RCA (Tiger Catheter, Terumo Co.). In case the diagnostic CAG did not reveal substantial stenosis (≥50% diameter stenosis on visual estimation), an intracoronary ergonovine injection was administered to induce coronary spasms. First a right-sided spasm provocation was performed together with a right-sided CAG, and if a spasm was induced, an intracoronary nitroglycerin injection was administered for relief. If the right-sided provocation test did not induce a coronary spasm, a left-sided provocation test was attempted. We injected 10–20 μg of ergonovine thrice at 1 min intervals into each coronary artery. Even in the negative cases, oral and intracoronary nitroglycerin (100 μg) and nifedipine (10 mg) were administered before completing the procedure to prevent delayed coronary spasms. A positive ergonovine test, which was defined as total or subtotal occlusion (visually, >90% stenosis) compared to the dilated condition after nitroglycerin administration, was validated by an electrocardiogram shift (>2 mm ST depression or elevation) and/or the presence of concomitant chest pain [19].

## CCTA acquisition and analysis

All CCTAs were conducted using a 320-detector row CT system (Aquilion One; Canon Medical Systems, Otawara, Japan) with two collimations of 320 × 0.5 mm, a gantry rotation time of 350 ms, and a temporal resolution of 175 ms. The protocol used for dual acquisition CT was according to the NAVIGATOR study [9,16]. A commercial software package (Sure Exposure 3DⓇ, Canon Medical Systems) was used to control the tube voltage (120 kVp) and tube current (130 to 250 mA). A bolus of 50–70 mL of nonionic contrast material (iobitridol, Xenetix Ⓡ 350 mg/mL; Guerbet, France) was intravenously infused at 4 mL/s, followed by infusion of 30 mL of a contrast/saline mixture (2:8 dilution) at 4 mL/s. The CT scans began with a 5-s delay after an automated bolus trigger in the ascending aorta (the triggering threshold was 100 Hounsfield units, HU). We did not administer additional beta-blockers or calcium channel blockers for decreasing the heart rate. All datasets were handled through iterative reconstruction (AIDR 3D, Canon Medical Systems). Axial images were reconstructed at a 0.5 mm slice thickness and 0.5 mm intervals in a field of view tailored to each patient's heart size.

Regarding per-vessel analysis, positive spasm vessels were determined by comparing the baseline and IV nitrate CTs. The three main coronary branches (RCA, left anterior descending artery (LAD), and left circumflex artery (LCX)) with luminal diameters >1.5 mm were analyzed. For both the CTs, we evaluated curved multiplanar reconstruction images and cross-sectional images of each coronary artery. For the same patient, we attempted to compare baseline and IV nitrate CT images of the coronary arteries in similar cardiac phases. Regarding VA diagnosis, the following were the requirements for a positive finding on CCTA: (a) significant focal stenosis with negative remodeling on baseline CT without definite evidence of plaques in a completely dilated artery on IV nitrate CT ("focal-type"), or (b) diffuse small diameter (<2 mm) of a major coronary artery with lack of tapering and beaded appearance on baseline CT that showed complete dilation on IV nitrate CT ("diffuse-type") (Figs 2 and 3) [10].

All CCTA images were independently reviewed by two radiologists (E.J.K and H.S.Y) who were blinded to the patients' clinical information, and discrepancies in results were resolved through consensus.

## TAG and proximal enhancement measurements

All images were analyzed using commercial software (VitreaⓇ, Vital images, MN, USA). TAG values were measured using semi-automated methods on dedicated computer software (Canon Medical Systems) for each of the three major epicardial coronary arteries (RCA, LAD, and LCX) as previously reported [20]. The centerline and contouring of each major coronary

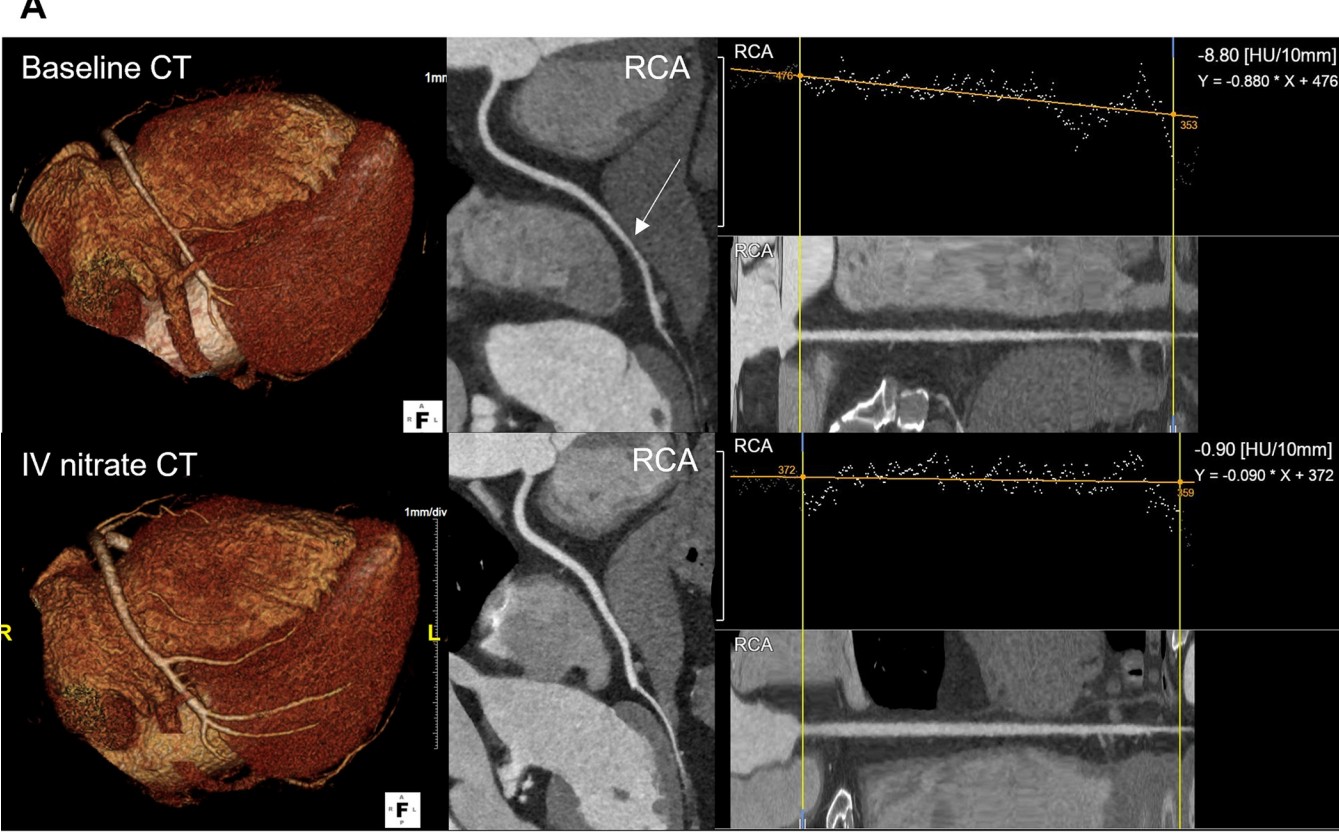

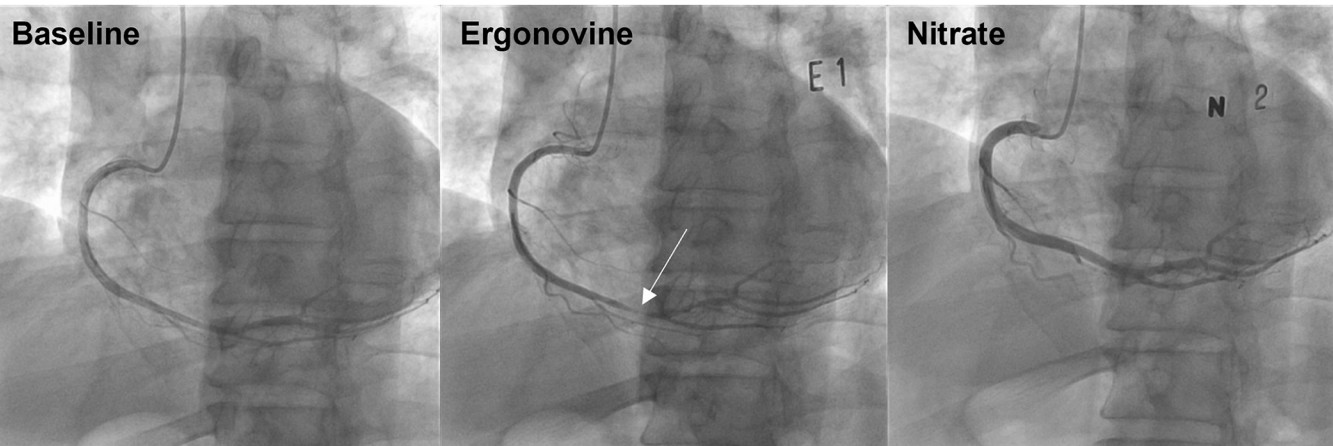

**Fig 2. A representative case of the focal-type spasm on CCTA and the ergonovine provocation test.** A 59-year-old male patient presented with intractable recurrent chest pain. (A) Baseline CT (upper) revealed significant focal stenosis at the distal RCA on volume rendering and curved multiplanar images. IV nitrate CT (lower) revealed completely dilated coronary arteries without narrowing. (B) Baseline coronary angiography (left) revealed intermediate luminal stenosis of the distal RCA. Upon ergonovine infusion (middle), RCA showed complete luminal occlusion. After nitrate injection, RCA showed full dilation without evidence of a stenotic lesion (right). The TAG values were -8.80 and -0.90 for baseline CT and IV nitrate CT, respectively. CCTA, coronary computed tomography angiography; CT, computed tomography; IV, intravenous; RCA, right coronary artery; TAG, transluminal attenuation gradient.

artery were automatically identified and manually modified, if necessary. Cross-sectional images perpendicular to the vessels' center-line were reconstructed. The mean luminal attenuation (HU) was measured at 1 mm intervals, from the ostium to distal levels where the cross-

sectional minimal area fell below 2 mm$^2$ [12]. Datapoints in segments with motion or blooming artifacts from luminal calcium were excluded when calculating TAG values [21]. TAG was defined as the linear regression coefficient between the intraluminal HU and the distance from the ostium. TAG values were calculated based on the change in CT attenuation (HU) per 10 mm length of the coronary artery. Representative examples are shown in Figs 2 and 3. Additionally, we assessed the contrast enhancement in the proximal ostium of the three major coronary arteries ("ProxHU") by drawing a region of interest as broad as possible while carefully avoiding calcifications in the cross-sectional images of each vessel's curved multiplanar planes.

## Statistical analysis

We performed group comparisons of the clinical features. Moreover, the mean TAG and ProxHU values for each major coronary artery (LAD, LCX, and RCA) were compared according to the CT acquisition method (baseline or IV nitrate CT), presence of spasm (spasm(-) vs. spasm(+)), and spasm types (focal vs. diffuse). Continuous and categorical variables are shown as means with standard deviations and frequencies (percentages), respectively. Between-group comparisons were performed using the independent t-test, Mann Whitney U test, and paired t-test, as appropriate. One-way analysis of variance was used to test within-group differences based on the normality of the data distribution. Pearson's correlation was used to examine the correlations between the TAG and ProxHU. Correlation coefficients of $< 0.20$, 0.20–0.39, 0.40–0.59, 0.60–0.79, and $\geq 0.80$ indicate very weak, weak, moderate, strong, and very strong correlations, respectively. Interobserver agreement for decisions of coronary spasm was assessed using a kappa test. All statistical analyses were performed using SPSS (version 20.0; SPSS Inc., Chicago, IL, USA). Statistical significance was set at $P < 0.05$.

## Results

### Clinical characteristics

The mean age of the study population was 60.67±10.05 years, and 72.1% of the patients were male (31/43; Table 1). There were 11, 3, 9 and 21 patients with hypertension, diabetes, dyslipidemia, and smoking history, respectively. Among the 43 recruited patients, coronary vasospasm was found in 24 patients (55.8%) while 19 (44.2%) showed negative results of coronary spasm. Compared with the spasm(-) group, the spasm(+) group had a significantly larger proportion of males (87.5% vs 52.6%, $P = 0.017$) and patients with a smoking history (66.7% vs 26.3%, $P = 0.011$); however, there were no significant between-group differences with regards to age and other comorbidities. There were no significant between-group differences in the height, weight, body mass index, and mean coronary arterial calcium score (Agatston method); however, we only enrolled patients without significant luminal stenosis as revealed by conventional CAG.

### Per-vessel spasm analysis using CCTA

Among the 24 patients with coronary vasospasm, 38 vessels showed coronary vasospasm (Table 2, Fig 4). Specifically, 10 patients had >2 vessels with spasm (one vessel, 14 patients; two vessels, 6 patients; three vessels, 4 patients). The most common location of vessels with spasm was LAD, followed by RCA and LCX (LAD, n = 21; RCA, n = 10; LCX, n = 7). For the subtype analysis of coronary spasm, the diffuse-type (n = 29, 76%) was more common than the focal-type (n = 9, 24%). In LAD, LCX, and RCA, 14 (33%), 7 (16%), and 8 (18%) vessels showed the diffuse type, respectively, while the corresponding values for the focal type were 7 (16%), 0 (0%), and 2 (5%) vessels, respectively. The interobserver agreements (weighted

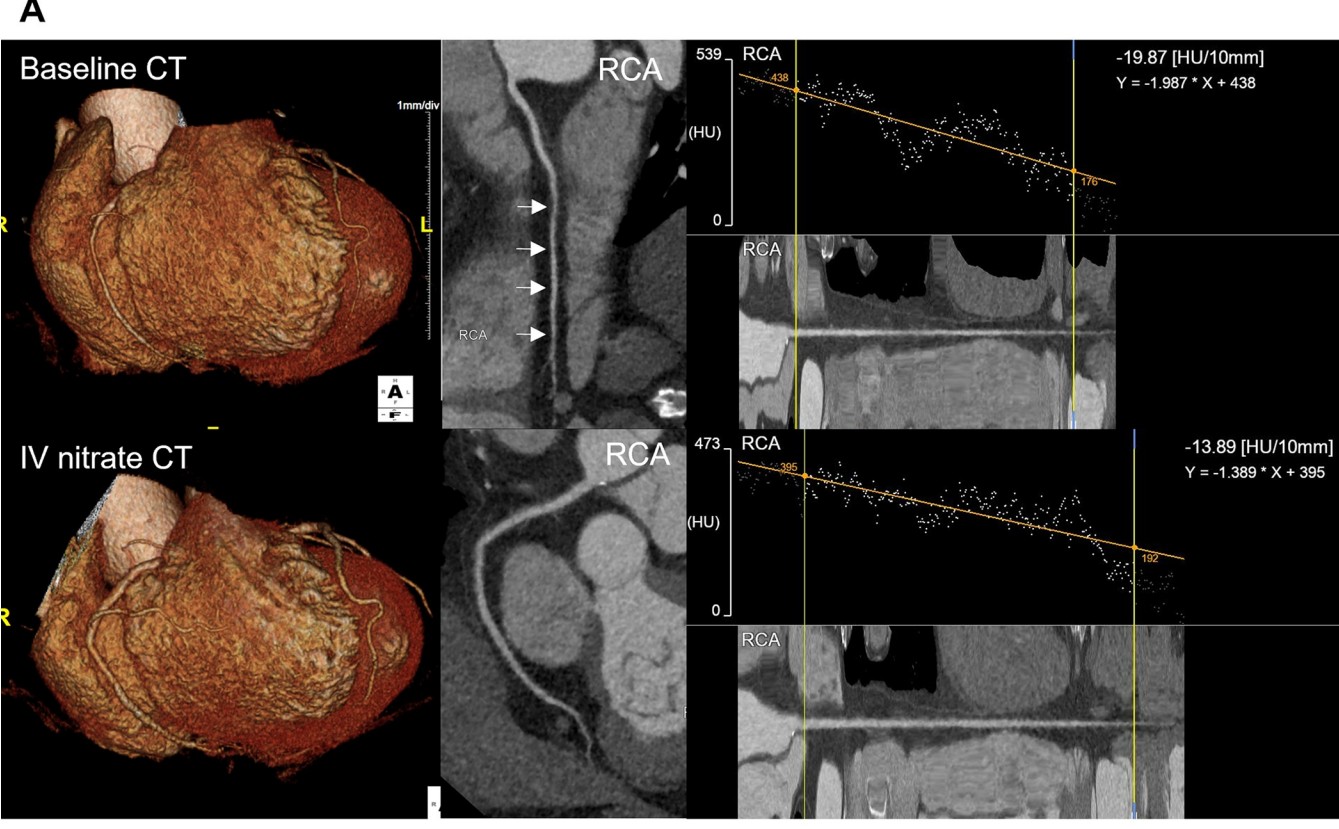

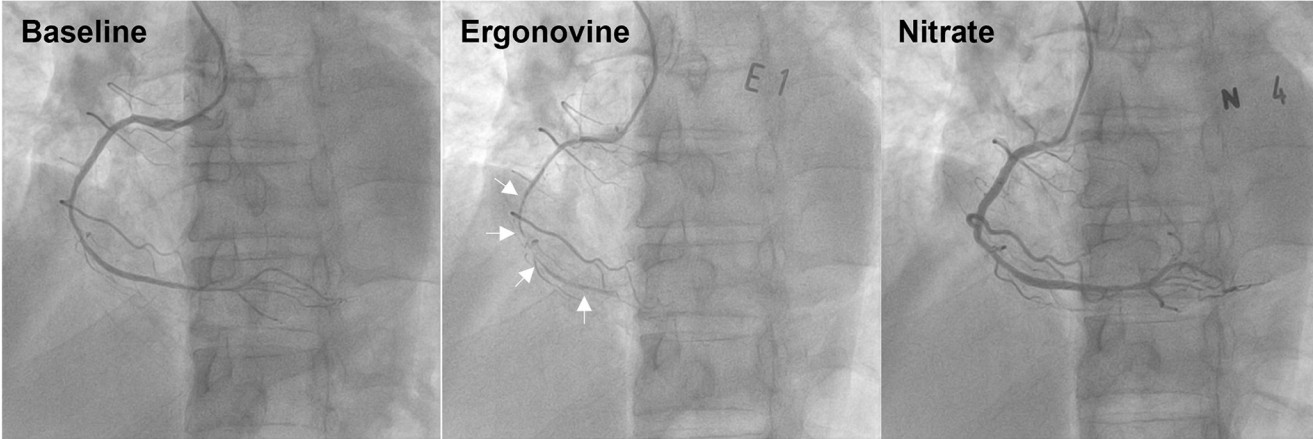

**Fig 3. A representative case of the diffuse-type spasm on CCTA and the ergonovine provocation test.** A 58-year-old male patient presented with chest pain. (A) Baseline CT (upper) revealed a diffuse small diameter with a beaded appearance throughout the coronary arteries at the distal RCA on volume rendering and curved multiplanar images. IV nitrate CT (lower) showed completely dilated coronary arteries without narrowing. (B) Baseline coronary angiography (left) demonstrated diffuse narrowing of the whole coronary branches, which was consistent with the findings on baseline CCTA (A). The TAG values were -19.87 and -13.89 on baseline CT and IV nitrate CT, respectively. CCTA, coronary computed tomography angiography; CT, computed tomography; IV, intravenous; RCA, right coronary artery; TAG, transluminal attenuation gradient.

kappa) between the radiologists regarding the decision of coronary spasm were 0.781 (95% confidence interval (CI): 0.601 to 0.960) per patient and 0.759 (95% CI: 0.618 to 0.901) per vessel.

**Table 1. General characteristics of the enrolled subjects.**

|  | Total | Spasm (–) | Spasm (+) | P |
|---|---|---|---|---|
| No. of patients | 43 | 19 | 24 |  |
| Age (years) | 60.67 ± 10.05 | 61.42 ± 13.56 | 60.08 ± 6.32 | 0.67 |
| Male sex (%) | 31 (72.1) | 10 (52.6) | 21 (87.5) | 0.017 |
| Hypertension (%) [a] | 11 (25.6) | 2 (10.5) | 9 (37.5) | 0.077 |
| Diabetes mellitus (%) [b] | 3 (7.0) | 1 (5.3) | 2 (8.3) | 1 |
| Dyslipidemia [c] | 9 (20.9) | 5 (26.3) | 4 (16.7) | 0.477 |
| Smoking history (%) |  |  |  | 0.011 |
| Never | 22 (51.2) | 14 (73.7) | 8 (33.3) |  |
| Current/Past | 21 (48.8) | 5 (26.3) | 16 (66.7) |  |
| Height (cm) | 164.95 ± 8.43 | 164.21 ± 7.28 | 165.54 ± 9.35 | 0.613 |
| Weight (kg) | 65.32 ± 9.78 | 64.37 ± 9.96 | 66.08 ± 9.78 | 0.576 |
| BMI (kg/m$^2$) | 23.95 ± 2.58 | 23.81 ± 2.61 | 24.06 ± 2.61 | 0.756 |
| Agatston score | 114.91 ± 223.67 | 159.15 ± 295.06 | 79.89 ± 142.57 | 0.253 |

Data are expressed as mean ± standard deviation or numbers of patients (%).

[a] Patients were considered hypertensive if their blood pressure was consistently > 140/90 mm Hg, or if they were currently taking anti-hypertensive medication.

[b] Patients were considered to have diabetes mellitus if their fasting glucose level was ≥ 126 mg/dL in, at least one assessment, or if they were currently taking oral hypoglycemic agents or insulin.

[c] Patients were considered to have dyslipidemia if they presented a range of lipid abnormalities in combination: increased total cholesterol (> 200 mg/dL), low-density lipoprotein cholesterol (> 140 mg/dL), and triglyceride levels (> 150 mg/dL) or decreased high-density lipoprotein cholesterol (< 40 mg/dL). Spasm (-), patients without vessels showing spasm; Spasm (+), patients with vessels showing spasm; BMI, body mass index.

## Comparison of TAG and ProxHU values among vessel types and CT acquisition methods

The mean TAG values of each coronary artery showed significant differences between the baseline CT and IV nitrate CT (Table 3). In the baseline CT, RCA showed the highest TAG values (gentlest slope), followed by LAD and LCX (RCA, -11.50±7.08; LAD, -20.92±8.37; LCX, -29.56±17.37; P<0.001). In the IV nitrate CT, TAG values of each vessels showed a similar trend as the baseline CT group (P<0.001). There were no significant among-group differences in the ProxHU values among the three major coronary arteries for both the CT protocols (baseline CT group, P = 0.730; IV nitrate group, P = 0.795). The mean TAG values were significantly lower (steeper slop) for baseline CT than that for IV nitrate CT, especially for LAD and LCX (LAD, -20.92±8.37 vs. -16.71±6.90; P<0.001; LCX, -29.56±17.37 vs. -22.75±11.97; P = 0.001; RCA, -11.50±7.08 vs. -9.94±5.67; P = 0.068; Table 3). The ProxHU values were higher for IV nitrate CT than that for baseline CT for all three vessels (LAD, 410.70±73.60 vs

**Table 2. Per vessel analysis for coronary spasm on CCTA.**

| Vessels | All (n) | Spasm (–) | Spasm (+) |
|---|---|---|---|
| LAD | 43 | 22 (51.2%) | 21 (48.8%) |
| LCX | 43 | 36 (83.7%) | 7 (16.3%) |
| RCA | 43 | 33 (76.7%) | 10 (23.3%) |
| Total | 129 | 91 | 38 |

Data are expressed as numbers of vessels (%).

CCTA, coronary CT angiography; Spasm (-), patients without vessels showing spasm; Spasm (+), patients with vessels showing spasm; LAD, left anterior descending artery; LCX, left circumflex artery; RCA, right coronary artery.

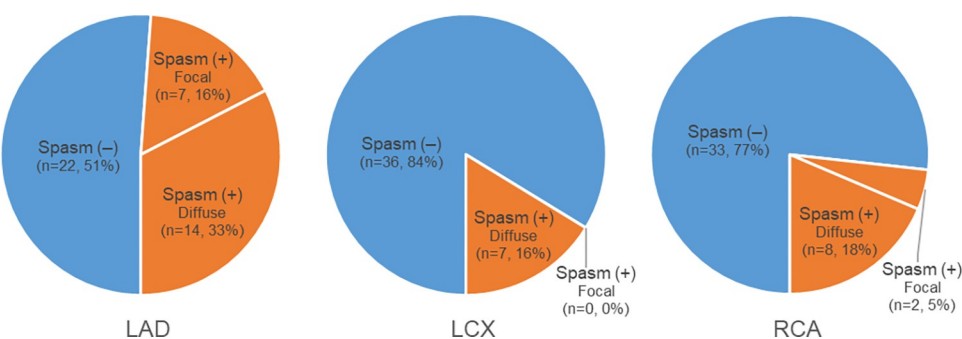

**Fig 4. Per vessel analysis of spasm type for coronary spasm on CCTA.** A total of 43 patients (129 vessels) were analyzed based on their spasm types. In LAD, LCX, and RCA, 14 (33%), 7 (16%), and 8 (18%) vessels showed the diffuse type, while the corresponding values for the focal type were 7 (16%), 0 (0%), and 2 (5%) vessels. LAD, left anterior descending artery; LCX, left circumflex artery; RCA, right coronary artery; Spasm (-), patients without vessels showing spasm; Spasm (+), patients with vessels showing spasm; Focal (type), significant focal stenosis without definite plaques; Diffuse (type), diffuse small diameter (< 2 mm) with serrated margin and loss of diameter tapering.

450.93±89.13; $P$ = 0.003; LCX, 407.11±62.82 vs 451.23±80.93; $P<$0.001; RCA, 399.11±73.41vs 440.20±90.21; $P$ = 0.004; Table 3).

ProxHU values were negatively correlated with the TAG values (baseline CT, $r$ = -0.360; $P<$0.001; IV nitrate CT, $r$ = -0.385; $P<$0.001; Fig 5). This trend was independent of the CT acquisition method.

## Comparison of TAG and ProxHU values between spasm(-) and spasm(+) vessels

In the baseline CT, the TAG and ProxHU values showed partial differences. Regarding TAG, LCX exhibited lower values in spasm(+) vessels than in spasm(-) vessels (-27.80±15.07 vs -44.62±22.04; $P$ = 0.016); however, there were no significant differences in TAG values for LAD and RCA (LAD, $P$ = 0.449; RCA, $P$ = 0.224, respectively; Fig 6). The ProxHU values of LAD exhibited lower values in spasm(+) vessels than in spasm(-) vessels (445.00±67.73 vs 375.00±64.79; $P$ = 0.001), while the other vessels did not show significant differences in baseline CT (LCX, $P$ = 0.579; RCA, $P$ = 0.532, respectively). In the IV nitrate CT, there were no

**Table 3. TAG and ProxHU values for the vessel types on baseline CT and IV nitrate CT.**

|  | Baseline CT | IV nitrate CT | *P* |
|---|---|---|---|
| TAG |  |  |  |
| LAD | -20.92±8.37 | -16.71±6.90 | <0.001 |
| LCX | -29.56±17.37 | -22.75±11.97 | 0.001 |
| RCA | -11.50±7.08 | -9.94±5.67 | 0.068 |
| *P* | <0.001 | <0.001 |  |
| ProxHU |  |  |  |
| LAD | 410.70±73.60 | 450.93±89.13 | 0.003 |
| LCX | 407.11±62.82 | 451.23±80.93 | <0.001 |
| RCA | 399.11±73.41 | 440.20±90.21 | 0.004 |
| *P* | 0.730 | 0.795 |  |

Data are expressed as mean ± standard deviation.

TAG, transluminal attenuation gradient; ProxHU, most proximal CT number of each coronary arteries; LAD, left anterior descending artery; LCX, left circumflex artery; RCA, right coronary artery.

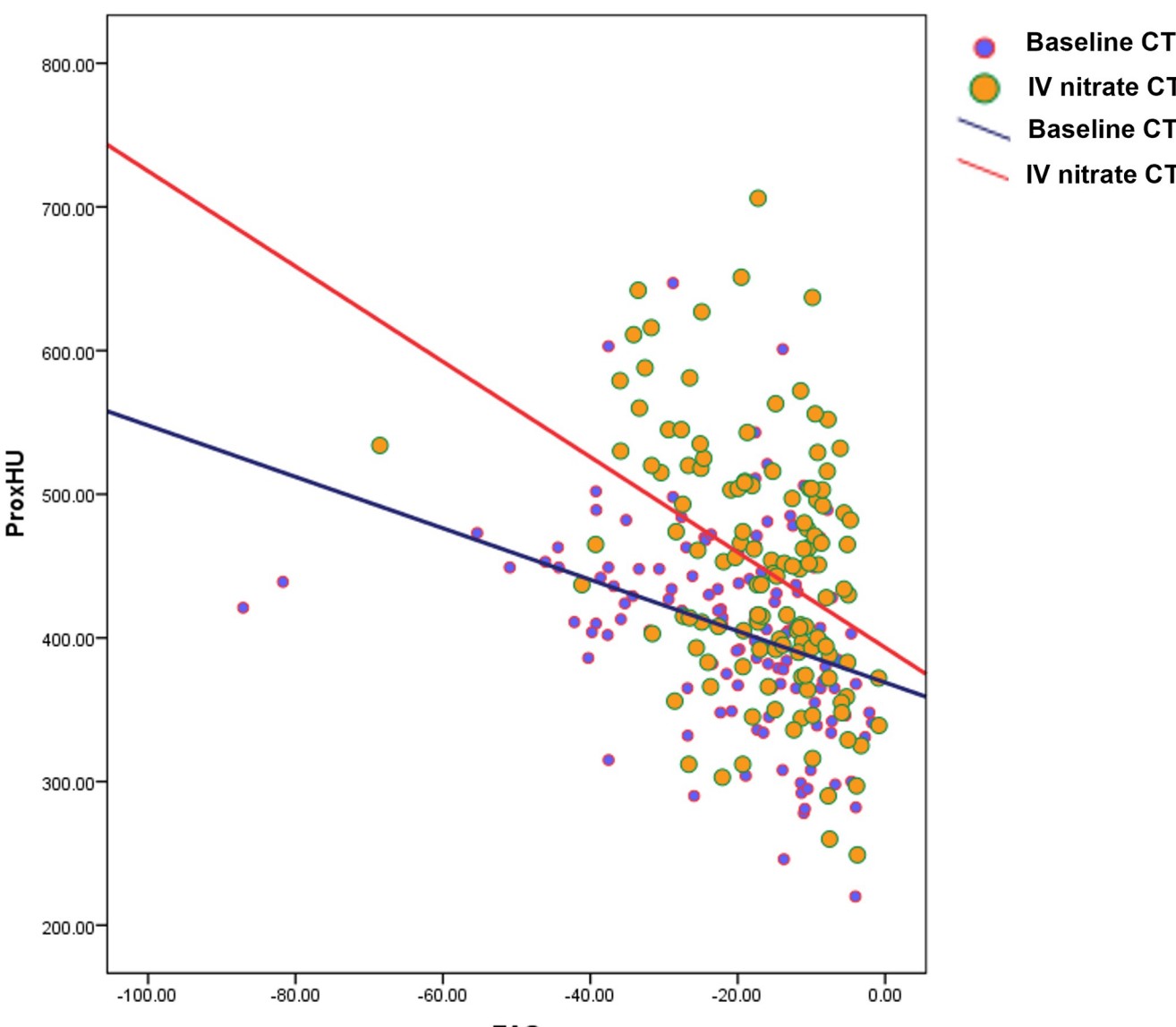

**Fig 5. Correlation between the TAG and ProxHU values of coronary arteries.** ProxHU values showed a weak negative correlation with TAG values (Baseline CT, *r* = -0.360; *P*<0.001; IV nitrate CT, *r* = -0.385; *P*<0.001). This trend was observed regardless of the CT acquisition method.

significant differences in either the TAG and ProxHU values between spasm(-) and (+) vessels for all the three major coronary arteries (*P*>0.05).

## Subgroup analysis of spasm(+) vessels on baseline CT

We performed subgroup analysis for spasm(+) vessels according to spasm type. Diffuse spasms showed significantly lower TAG values than focal spasms on the LAD and RCA (LAD, -14.02 ±4.49 vs. -23.02±10.45; *P* = 0.044; RCA, -8.77±0.04 vs. -15.22±6.62; *P* = 0.037; Table 4). There was no significant difference in the ProxHU values between the spasm types for the LAD and RCA. For LCX, all spasm(+) vessels were diffuse-type (TAG, -44.62±22.04); moreover, they showed the lowest TAG values compared with the other vessels.

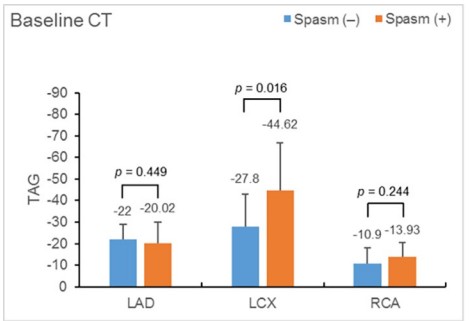
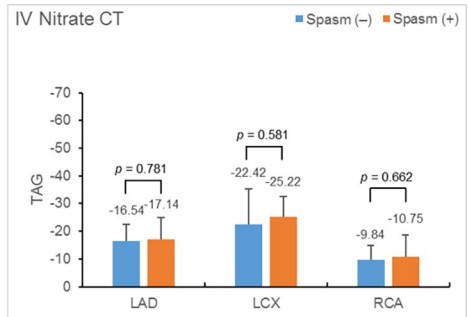
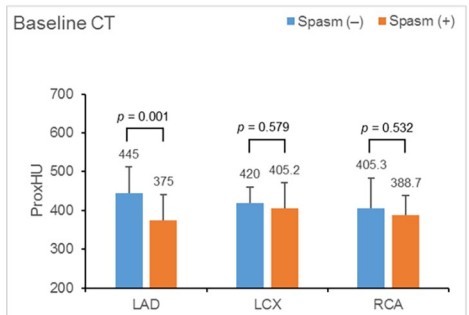
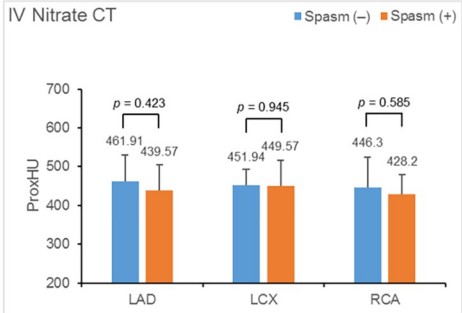

**Fig 6.** Comparison of TAG and ProxHU values between spasm(-) and spasm(+) vessels in baseline CT (A, B) and IV nitrate CT (C, D). In the baseline CT, the TAG of LCX was significantly lower values in spasm(+) vessels than in spasm (-) vessels; however there was no significant difference for that of LAD and RCA. The ProxHU values of LAD exhibited lower values in spasm(+) vessels than in spasm(-) vessels, while the other vessels did not show significant differences in baseline CT. For IV nitrate CT, there was no significant difference in the TAG or ProxHU values between the spasm (+) and (-) vessels for all three vessels. Spasm (-), patients without vessels showing spasm; Spasm (+), patients with vessels showing spasm; TAG, transluminal attenuation gradient; ProxHU, most proximal CT number of each coronary arteries; LAD, left anterior descending artery; LCX, left circumflex artery; RCA, right coronary artery.

## Comparison of intra-subject differences in TAG values between baseline CT and IV nitrate CT

The intra-subject difference in TAG value between the baseline CT and IV nitrate CT shows relatively larger value in diffuse type spasm vessels than in focal type spasm or negative spasm

**Table 4. Subgroup analysis of spasm (+) vessels on baseline CT.**

|  | Total spasm (+) | Focal type | Diffuse type | *P* |
|---|---|---|---|---|
| TAG |  |  |  |  |
| LAD | -20.02±9.79 | -14.02±4.49 | -23.02±10.45 | 0.044 |
| LCX | -44.62±22.04 | - | -44.62±22.04 | - |
| RCA | -13.93±6.44 | -8.77±0.04 | -15.22±6.62 | 0.037 |
| ProxHU |  |  |  |  |
| LAD | 375.00±64.79 | 388.00±49.80 | 368.50±71.95 | 0.529 |
| LCX | 405.19±67.18 | - | 405.19±67.18 | - |
| RCA | 388.70±50.73 | 417.50±74.25 | 381.50±47.17 | 0.428 |

Data are expressed as mean ± standard deviation.

TAG, transluminal attenuation gradient; ProxHU, most proximal CT number of each coronary arteries; Spasm (+), patients with vessels showing spasm; LAD, left anterior descending artery; LCX, left circumflex artery; RCA, right coronary artery; Focal type, significant focal stenosis without definite plaques; Diffuse type, diffuse small diameter (< 2 mm) with serrated margin and loss of diameter tapering.

**Table 5. Comparison of intra-subject differences in TAG values among normal, focal and diffuse spasm (+) vessels.**

|  | Spasm (-) | Spasm (+), Focal type | Spasm (+), Diffuse type | *P* |
|---|---|---|---|---|
| TAG difference [a] |  |  |  |  |
| Total | 3.83±7.04 | 2.05±6.13 | 6.20±13.77 | 0.35 |
| LAD | 5.46±7.60 | 0.93±6.50 | 3.86±6.00 | 0.326 |
| LCX | 5.38±7.07 | - | 15.11±25.68 | 0.052 |
| RCA | 1.07±5.88 | 5.97±2.74 | 2.48±4.52 | 0.433 |

Data are expressed as mean ± standard deviation.

TAG difference

[a], subtraction of TAG value of baseline CT from TAG value of IV Nitrate CT; TAG, transluminal attenuation gradient; Spasm (+), patients with vessels showing spasm;

LAD, left anterior descending artery; LCX, left circumflex artery; RCA, right coronary artery; Focal type, significant focal stenosis without definite plaques; Diffuse type,

diffuse small diameter (< 2 mm) with serrated margin and loss of diameter tapering.

vessels, but statistically insignificant (diffuse type vs focal type vs normal, 6.20±13.77 vs 2.05 ±6.13 vs 3.83±7.04; P = 0.35; Table 5). Among vessel subtypes, LCX, of which all spasm(+) vessels were diffuse type, showed a greater TAG difference for diffuse spasm(+) than for normal type, although this was not significant (diffuse type vs normal, 15.11±25.68 vs 5.38±7.07; *P* = 0.052; Table 5). The other two vessels, LAD and RCA, did not show significant TAG difference values among normal, focal spasm, and diffuse spasm vessels (LAD, *P* = 0.326; RCA, *P* = 0.433; Table 5).

## Discussion

In our study, 55.8% (24/57) of the patients were diagnosed with coronary spasms based on the spasm provocation test. There were significant differences in TAG values among the major coronary arteries, regardless of the CT acquisition method, with RCA showing the highest TAG value (RCA > LAD > LCX). Further, TAG values were significantly lower (steeper) in the baseline CT than in the IV nitrate CT. More vessels showed diffuse spasms than focal spasms (76% vs. 24%), with diffuse spasms exhibiting significantly lower TAG values than the focal spasms.

The TAG value is positively correlated with the vessel diameter and length [12,22]. In our study, there were significant differences in the mean TAG values among the three major coronary arteries for both CT protocols. Additionally, the TAG values were progressively lower for LCX, LAD, and RCA, which corresponds to the order of increasing vessel diameter. Furthermore, administration of a vasodilator increased the TAG values in all three major coronary arteries. LAD and LCX demonstrated more significant increase in TAG values than RCA. Both the diameter and the contrast gradient over distance in the major coronary branches showed a more significant change in the left side than in the right side [23,24]. This could be attributed to the smaller change in the vessel diameter of the RCA over the vessel distance compared with the left coronary artery, which could result from the RCA having a larger diameter and longer length, as previously reported [25].

In our study, IV nitrate CT showed higher TAG (gentler slope) and ProxHU values than baseline CT in all three major coronary arteries. We previously found that the TAG and ProxHU values of coronary arteries increased with vasodilator administration on CCTA, regardless of the administration method (IV or sublingual), which is consistent with the present findings [25]. Additionally, there was a weak negative correlation between TAG and ProxHU values in all major coronary arteries before and after vasodilator administration.

Coronary arteries are dilated after vasodilator administration, which emphasizes the contrast enhancement effect of increasing TAG and ProxHU values [25,26].

Comparison of TAG values in baseline CT between spasm(-) and (+) vessels, only LCX exhibited significantly lower values in spasm(+) than in spasm (-), but not in LAD and RCA. Moreover, the intrasubject TAG difference in LCX was greater in diffuse spasm(+) than normal group. This could be attributed to the relatively hypoplastic nature of LCX, and the small LCX sample size which only included diffuse spasm(+) vessels.

According to conventional CAG-based studies, coronary spasm can be divided into two types, based on the spasm length on CAG induced by the provocation test as follows: "focal spasm", i.e., vasoconstriction within one coronary segment, and "diffuse spasm", i.e., vasoconstriction of more than one adjacent coronary segments [27,28]. It remains unclear whether there are differences in the prognosis and clinical course between the spasm types. Sato et al. [29] analyzed 5-year clinical outcomes and showed that diffuse spasms showed fewer major adverse cardiovascular events and a better prognosis than focal spasms. Conversely, Park et al. [30] analyzed 3 year clinical outcomes and found that diffuse coronary spasms were independent predictors of recurrent chest pain. Sueda et al. [31] suggested that diffuse spasms had poor responses to medical treatment. Akasaka et al. [32] reported that vessels with diffuse spasms had significantly lower coronary flow reserves than those with focal spasms, which suggested that focal spasms are associated with localized endothelial dysfunction of the epicardial coronaries with no significant effect on coronary microvascular function. This explains the better prognosis of focal-type spasms. However, most CCTA studies on coronary spasms only considered focal spasms as a morphologic feature [8,10,33]. Due to the significant individual differences in the vessel diameter and density on contrast enhancement, as well as the influence of the contrast injection protocol or clinical characteristics, it might be difficult to identify diffuse type spasms on conventional single-acquisition CCTA. Using the double-acquisition protocol, we could assess diffuse spasms and determine the spasm types on CCTA. Moreover, for spasm(+) LAD and RCA vessels, all diffuse-type vessels showed significantly lower TAG values than the focal-type vessels. Since TAG is positively correlated with the diameter [12], the relatively small diameters of diffuse-type spasm vessels could have resulted in lower TAG values as compared to those of focal-type spasm vessels which usually maintain a normal diameter from the distal to spasm site. We cannot conclusively determine whether the vessel diameter is the only factor influencing TAG in diffuse-type spasm vessels or that it may co-exist with other factors, including coronary microvascular function status. There is a need for further studies on the poor prognosis and low TAG values of diffuse spasms, to improve VA diagnosis and management.

In our previous study [9], we examined the feasibility of the double acquisition CCTA protocol, which showed a relatively higher sensitivity for diagnosing VA compared with previous conventional single acquisition CCTA protocols. Moreover, according to the present study, the differences in TAG values of coronary spasms depend on the morphological subtype, which may improve the detection of diffuse type spasm. However, due to numerous limitations, including the difficulty of performing CCTA twice for the same patient, variations in vasospasm timing, increased radiation exposure, increased amount of contrast material used, and the requirement of CAG for confirmatory diagnosis of VA, further research is needed to determine the clinical use of the double acquisition CCTA protocol.

This study had several limitations. First, we included a relatively small sample size; therefore, future large-scale prospective studies are needed to confirm our results. Second, we excluded patients with significant (>50%) fixed stenosis of the coronary artery. Stenotic vessels have lower TAG values than non-stenotic vessels [21], which suggests that the inclusion of stenotic vessels may have influenced our results. Nonetheless, given the difficulty of

discriminating between stenotic and spastic vessels, we chose to exclude fixed stenotic vessels. Third, we excluded 6 patients with inconsistent results on CCTA (negative) and the ergonovine provocation test (positive) in the analysis of TAG values. These false-negative results could result from variations in vasospasm timing and its migratory nature, which make them unreliable for determining the existence of vasospasm at the time of CCTA acquisition. Thus, due to the possible effect of inconsistencies between CCTA and CAG in the TAG analysis, we only included concordant resulting vessels to achieve accurate TAG results. Fourth, the results regarding the TAG values in spasm(+) and (-) vessels, regardless of vessel and spasm types, could not support an improved VA diagnostic performance with the addition of TAG in the CCTA analysis. We therefore focused on presenting differences in the TAG values between the spasm types.

In conclusion, coronary spasms can be classified as diffuse and focal types on CCTA. Additionally, a relatively large proportion of coronary spasms present as diffuse spasms rather than focal spasms. The TAG values of coronary spasms significantly differed according to the morphological feature. Diffuse-type spasms showed significantly lower TAG values than focal-type spasms. There is a need for future large-scale prospective studies to reveal the diagnostic utility of TAG in discriminating coronary spasms on CCTA.

## Supporting information

**S1 Table. Per-vessel analysis of diagnostic performance of CCTA.**
(DOCX)

**S2 Table. Minimal data set for Tables 1–5 and S1 and Figs 1–6.**
(XLSX)

## Author Contributions

**Conceptualization:** Eun-Ju Kang, Moo Hyun Kim.

**Data curation:** Jae Yang Park, Eun-Ju Kang, Moo Hyun Kim, Hwan Seok Yong, Seung-Woon Rha.

**Formal analysis:** Jae Yang Park, Eun-Ju Kang.

**Funding acquisition:** Eun-Ju Kang.

**Investigation:** Eun-Ju Kang.

**Methodology:** Eun-Ju Kang.

**Project administration:** Eun-Ju Kang.

**Resources:** Eun-Ju Kang.

**Software:** Eun-Ju Kang.

**Supervision:** Eun-Ju Kang, Moo Hyun Kim.

**Validation:** Eun-Ju Kang.

**Visualization:** Eun-Ju Kang.

**Writing – original draft:** Jae Yang Park.

**Writing – review & editing:** Jae Yang Park, Eun-Ju Kang.

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
