## [Decision Letter · Decision Letter 0]

9 Feb 2022

PONE-D-21-39960Assessment of coronary spasms with transluminal attenuation gradient in coronary computed tomography angiographyPLOS ONE

Dear Dr. Kang,

Thank you for submitting your manuscript to PLOS ONE. It was reviewed by three reviewer's and all agreed that the manuscript does not meet PLOS ONE’s publication criteria as it currently stands. While the reviewers felt that the focus of the manuscript was interesting, they differed widely on the study's methodologic soundness. If you believe that you can adequately address all of the reviewer's concerns we do invite you to submit a revised manuscript. The Editor is in agreement with the sentiment that the impact of the manuscript would be significantly improved by a more thorough analysis of spasm detected non-invasively by transluminal attenuation gradient to that detected by invasive provocative testing. Given the extensive changes that are recommended, please note that a revised manuscript may be sent to additional reviewers and there is no guarantee of final acceptance.  Please submit your revised manuscript by Mar 26 2022 11:59PM. We realize that extensive revisions may require more time to complete. If this is the case, please reply to this message or contact the journal office at plosone@plos.org. Please include the following items when submitting your revised manuscript:A rebuttal letter that responds to each point raised by the academic editor and reviewer(s). You should upload this letter as a separate file labeled 'Response to Reviewers'.A marked-up copy of your manuscript that highlights changes made to the original version. You should upload this as a separate file labeled 'Revised Manuscript with Track Changes'.An unmarked version of your revised paper without tracked changes. You should upload this as a separate file labeled 'Manuscript'.

We look forward to receiving your revised manuscript.

Kind regards,

Jeffrey J. Rade, MD

Academic Editor

PLOS ONE

Journal Requirements:

2.  Thank you for submitting the above manuscript to PLOS ONE. During our internal evaluation of the manuscript, we found significant text overlap between your submission and the following previously published works, some of which you are an author.

- https://epos.myesr.org/poster/esr/ecr2020/C-06177/Results

- https://kjronline.org/DOIx.php?id=10.3348%2Fkjr.2019.0908

- http://jcr-new.cjint.kr/!Speaker/upload/1416790567_phpMt5RM9_-2.pdf

Please revise the manuscript to rephrase the duplicated text, cite your sources, and provide details as to how the current manuscript advances on previous work. Please note that further consideration is dependent on the submission of a manuscript that addresses these concerns about the overlap in text with published work.

Reviewers' comments:

Reviewer's Responses to Questions

**Comments to the Author**

1. Is the manuscript technically sound, and do the data support the conclusions?

Reviewer #1: Yes

Reviewer #2: Yes

Reviewer #3: Partly

2. Has the statistical analysis been performed appropriately and rigorously? 

Reviewer #1: Yes

Reviewer #2: Yes

Reviewer #3: No

3. Have the authors made all data underlying the findings in their manuscript fully available?

Reviewer #1: Yes

Reviewer #2: Yes

Reviewer #3: No

4. Is the manuscript presented in an intelligible fashion and written in standard English?

Reviewer #1: Yes

Reviewer #2: Yes

Reviewer #3: No

5. Review Comments to the Author

Reviewer #1: This manuscript is a well conceived and executed extension of prior work on the topic of using TAG in CCTA. It provides a potential novel tool for measuring/evaluating coronary vasospasm on CCTA. Below are specific comments and recommendations for revision:

1. Relatively high “false positive VA” (8/57 excluded for fixed stenoses) and “false negative VA” (6/57 excluded for negative CCTA but positive conventional angiogram) in the initial study population. The patients with fixed stenoses are addressed as the second limitation of the study. Please address and account for the false negative patients. These findings suggest that there is significant clinical overlap within the variant chest pain population.

2. For institutions that perform only CCTA with vasodilator, vasospasm will likely go undiagnosed. Would you recommend adding baseline CT to all variant chest pain CCT protocols in order to help diagnose VA?

3. Because hypertension and smoking are so significantly correlated with positive vasospasm patients, is TAG more useful in that population? Please comment on this in the discussion.

4. Nondiagnostic coronary segments were manually removed from the analysis (i.e, due to motion or blooming artifact) Can you include details about how many times was this necessary? Were there any entirely nondiagnostic CCTA exams?

5. Who performed the CCTA analysis. Was it one or more than one radiologist? If more than one, were there are any discrepant results and how were those adjudicated.

6. In the discussion, do you mean: We observed that IV nitrate CT shows (?higher?) TAG and higher ProxHU values than baseline CT in all three major coronary arteries. Can you clarify?

Reviewer #2: This is a very interesting paper with very good angiographic and CT correlation in patients with coronary vasospasm. I have only a few comments. Please see below.

1. Lines 71-75. I don't understand these sentences. I think the introduction warrants minor expansion to clarify the method of TAG measurement. Also, to explain the influence of vessel diameter on TAG measurement.

2. Line 140. Were beta blocker or calcium channel blockers used to lower heart rate for CT scans? Was there a difference in the use of these medications in the spasm and non spasm patients? Was the mean heart rate at the time of CT scan the same in both groups (with and without NTG?).

3. Line 121 and 122. I believe that the character "u" was omitted from the text. The text reads "10-20 g of ergonovine" and "nitroglycerin 100g". Perhaps simply use the abbreviation "ucg" or consult with the editorial staff regarding the use of greek characters.

4. Line 162. I assume that the analysis of TAG was terminated at the distal RCA prior to the bifurcation of the PDA and PLV branch. IF so, this should be clarified in the methods.

5. Line 185. I believe "distal levels" should be "distal vessels".

6. Line 188. I believe "ostium length" should be changed to "vessel length".

7. Line 266. Is there a minus sign missing from the baseline LAD TAG value?

8. Was there a difference in coronary dominance between spasm positive and spasm negative patients?

9. Figure 5. I believe these results are also reported in table 4. I would consider omitting table 4, and only use the figure. If you do this, you need to label the results of statistical analysis in the figure.

10. Figure 5. Panel B is incorrectly labeled as IV nitrate CT.

11. Figure 5. Panel C is incorrectly labeled as baseline CT.

12. Line 364, discussion. Please change "epithelial" to "endothelial".

Reviewer #3: The non-invasive diagnosis of patients with vasospastic angina is a worthy goal. This manuscript describes the imaging features, mainly transluminal attenuation gradient (TAG) and contrast enhancement of the proximal ostium (ProxHU), using dual (before and after iv nitroglycerin) coronary CT angiography (CCTA) of 43 enrolled in the NAVIGATOR study who had a high likelihood of vasospastic angina. 24 of 43 subjects of enrolled subjects had vasospasm on provocative invasive angiography. Of the 72 major coronary vessels in these 24 subjects, 76% demonstrated diffuse spasm on CCTA while 26% had discrete spasm. As expected, TAG was numerically less negative (i.e. less steep) and ProxHU greater in major vessels after administration of iv NTG (Table 3). The difference in TAG between vessels with and without spasm was only significant on the baseline was in the left circumflex artery (which was affected least often by spam as indicated in Table 1) and the only difference in ProxHU was observed in LAD. Not unexpectedly, there were no difference in vessels with and without spam after iv NTG administration. Also not unexpectedly, diffuse spasm was associated with a more negative TAG and lower ProxHU than discrete spasm. The authors conclude that a relatively large percentage of coronary spasms present as diffuse and TAG significantly differed according to the morphological type of the coronary spasm.

The concept of measuring TAG by CCTA before and after iv NRG administration to identify patients with vasospastic angina is intriguing and a potentially useful proposition. Unfortunately, the singularly descriptive and unsophisticated nature of the data presented in this manuscript does little to advance this aim. The authors have an opportunity to go well beyond that with their data set and determine if the change in TAG with iv NTG, both on a per patient and per vessel basis, correlates with vasospasm detected on the gold standard invasive provocative test. The impact of the manuscript would be far greater if they did. In addition, there are several other major criticisms of the manuscript. These include:

1. Subjects for this analysis were enrolled in the NAVIGATOR study but according to the references for that study, only 41 subjects with vasospastic angina were enrolled. The authors need to clarify the discrepancy with the reported 57 enrolled subjects. During what dates were subjects enrolled in the NAVIGATOR study?

2. The authors excluded 6 subjects with invasive evidence of vasospasm in whom the CCTA did not reveal spasm (false negative). This is a major source of bias, and these subjects should be included in the analysis.

3. Table 2 is incredibly confusing. I am assuming column 3 represents all vessels without and column 4 all vessels with spasm on CCTA. What one cannot tell from that table is how many vessels without spasm on invasive imaging had spasm on CCTA. If the goal of the table was to depict the proportion of spasm type (focal vs diffuse) in the specific arteries of patients with spasm detected by CCTA, a figure would likely be better. On a related note, what was the correlation spasm detected in the same arteries on invasive provocative testing and were the proportions of spasm type similar?

4. Table 3 shows mean TAG and ProxHU results and per vessel differences before and after iv NTG. Subjects with and without spasm on invasive testing are lumped together but would it not be more useful to determine if the change in each parameter with NTG is different between subjects with and without invasive evidence of spasm?

5. The information presented in Table 4 and Figure 4 are identical and therefore redundant. Furthermore, some of the same data is also presented in the text.

6. The manuscript would be greatly improved if the authors correlated TAG (or more specifically the change in TAG with iv NRG) to the gold standard invasive provocative testing. Calculation of sensitivity/specificity and positive/negative predictive value would help determine if measuring TAG (or ProxHU) by CCTA is a useful non-invasive method to diagnose vasospastic angina.

Minor comments:

1. The authors need to better clarify what type vasospasm to which they are referring. The reader finds it hard to know if they are talking about spasm detected by invasive provocative testing of by CCTA.

2. There are several grammatical errors that should be remedied.

6. PLOS authors have the option to publish the peer review history of their article (what does this mean?). If published, this will include your full peer review and any attached files.

Reviewer #1: No

Reviewer #2: **Yes: **Timothy P. Fitzgibbons

Reviewer #3: No

---

## [Author Response · Author response to Decision Letter 0]

24 Mar 2022

Response to reviewers

Reviewer #1: 

1. Relatively high “false positive VA” (8/57 excluded for fixed stenoses) and “false negative VA” (6/57 excluded for negative CCTA but positive conventional angiogram) in the initial study population. The patients with fixed stenoses are addressed as the second limitation of the study. Please address and account for the false negative patients. These findings suggest that there is significant clinical overlap within the variant chest pain population.

Answer: Thank you for this thoughtful advice. The six patients showed negative results on CCTA but positive results in the ergonovine provocation test. This could be attributed to individual differences and variations in the timing of CT acquisition. Given the variations in the timing and migratory nature of vasopasms, CCTA studies on coronary spasms have generally discarded such false negative results; accordingly, we exclude them from the results. 

Line 517-521; Third, we excluded patients with inconsistent results on CCTA (negative) and the ergonovine provocation test (positive). False-negative results could result from variations in the timing and migratory nature of vasospasms; moreover, they are generally excluded in related studies. 

2. For institutions that perform only CCTA with vasodilator, vasospasm will likely go undiagnosed. Would you recommend adding baseline CT to all variant chest pain CCT protocols in order to help diagnose VA?

Answer: Thank you for your suggestion. Although our double-acquisition CCTA protocol improved the sensitivity for diagnosing vasospastic angina compared with conventional single-acquisition CCTA protocols (sensitivity in per-patient, 73%; Eur Radiol. 2017;27: 1136–1147), it has several drawbacks in routine clinical practice (radiation dose, contrast media, etc.). Additionally, vasospasms show individual differences at various time during the day, which results in false negative results on CCTA. Moreover, since the spasm provoking test combined with invasive coronary angiography remains necessary for confirming variant angina, our double acquisition CCTA protocol cannot be used an independent diagnostic tool yet. 

Line 501-509; In our previous study (9), we examined the feasibility of the double acquisition CCTA protocol, which showed a relatively higher sensitivity for diagnosing VA compared with previous conventional single acquisition CCTA protocols. Moreover, according to present study, we found difference of TAG values of coronary spasms depend on the morphological subtype which may improving the detection of diffuse type spasm. However, due to numerous limitations, including the difficulty of performing CCTA twice for the same patient, variations in the time of vasospasm, increased radiation exposure, increased amount of contrast material used, and the requirement of CAG for confirmatory diagnosis of VA, further research is needed to determine the clinical use of the double acquisition CCTA protocol.

3. Because hypertension and smoking are so significantly correlated with positive vasospasm patients, is TAG more useful in that population? Please comment on this in the discussion.

Answer: Thank you for your suggestion. Male sex, smoking, and hypertension are established clinical risk factors for coronary spams, which is consistent with our findings. Given our small sample size and the limited number of patients with those comorbidities (hypertension 11/57, smoking 21/43), we could not analyze the actual correlations between the patients’ characteristics and TAG values. Moreover, we observed that the TAG value is significantly dependent on the type of coronary spasm. Therefore, we cannot guarantee the utility of additional TAG analysis in the detection of coronary spasms on CCTA. 

4. Nondiagnostic coronary segments were manually removed from the analysis (i.e, due to motion or blooming artifact) Can you include details about how many times was this necessary? Were there any entirely nondiagnostic CCTA exams?

Answer: Based on previous reports regarding TAG measurement, segments with motion or blooming artifacts are generally excluded when calculating the TAG values (JACC Cardiovasc Imaging. 2016;9(9):1074–83.; Radiology. 2016;279(1):75–83.); accordingly, we adapted the previously described methodology. Since these segments usually occupy only a small portion of the entire vessel, no vessels were excluded in this step. Therefore, all vessels were available for TAG analysis. We have revised the sentence for improved clarity. 

Line 231-232; Datapoints in segments with motion or blooming artifacts from luminal calcium were excluded when calculating TAG values (21). 

5. Who performed the CCTA analysis. Was it one or more than one radiologist? If more than one, were there are any discrepant results and how were those adjudicated.

Answer: Thank you for this thoughtful comment. In our study, two radiologists (EJ Kang, HS Yong) independently performed CCTA analysis, with discrepancies being reconciled through consensus. The weighted kappa for agreement between radiologist for decisions regarding coronary spasms were 0.781 (95% confidence interval (CI): 0.601 to 0.960) per patient and 0.759 (95% CI: 0.618 to 0.901) per vessel.

Line 218-220; All CCTA images were independently reviewed by two radiologists (E.J.K and H.S.Y) who were blinded to the patients’ clinical information, and discrepancies in results were resolved through consensus. 

Line 257-258; Interobserver agreement for decisions of coronary spasm was assessed using a kappa test. 

Line 303-306; The weighted kappa for agreement between the radiologists regarding decisions of coronary spasm were 0.781 (95% confidence interval (CI): 0.601 to 0.960) per patient and 0.759 (95% CI: 0.618 to 0.901) per vessel.

6. In the discussion, do you mean: We observed that IV nitrate CT shows (?higher?) TAG and higher ProxHU values than baseline CT in all three major coronary arteries. Can you clarify?

Answer : Thank you for pointing out this unclear point. In our study, the coronary arteries showed an increased TAG value (gentle slope) after vasodilator administration, which could be attributed to their dilatation emphasizing the contrast enhancement effects and increasing the TAG. We have revised this sentence for clarity.

Line 451-452; In our study, IV nitrate CT showed higher TAG (gentler slope) and ProxHU values than baseline CT in all three major coronary arteries. 

Reviewer #2: 

1. Lines 71-75. I don't understand these sentences. I think the introduction warrants minor expansion to clarify the method of TAG measurement. Also, to explain the influence of vessel diameter on TAG measurement.

Answer: Thank you for this thoughtful comment. We have modified this section for more clarity. 

Line 82-84; However, clinical validation studies have reported conflicting results of the usefulness of using TAG at determining coronary arterial flow (14,15), since TAG may be affected by changes in coronary luminal diameter and collateral vascular enhancement.

2. Line 140. Were beta blocker or calcium channel blockers used to lower heart rate for CT scans? Was there a difference in the use of these medications in the spasm and non spasm patients? Was the mean heart rate at the time of CT scan the same in both groups (with and without NTG?).

Answer: Thank you for your thoughtful comment. In our research, we did not administer beta blockers or calcium channel blockers during CT acquisition. There was no significant difference in the heart rate during CT acquisition between the CT protocols.

Line 173-174; We did not administer additional beta-blockers or calcium channel blockers for decreasing the heart rate. 

3. Line 121 and 122. I believe that the character "u" was omitted from the text. The text reads "10-20 g of ergonovine" and "nitroglycerin 100g". Perhaps simply use the abbreviation "ucg" or consult with the editorial staff regarding the use of greek characters.

Answer: Thank you for your careful observation. We have revised our manuscript accordingly.

Line 152-155; We injected 10–20 μg of ergonovine thrice at 1 min intervals into each coronary artery. Even in the negative cases, oral and intracoronary nitroglycerin (100 μg) and nifedipine (10 mg) were administered before completing the procedure to prevent delayed coronary spasms.

4. Line 162. I assume that the analysis of TAG was terminated at the distal RCA prior to the bifurcation of the PDA and PLV branch. IF so, this should be clarified in the methods.

Answer: Thank you for this thoughtful comment. We calculated the TAG of each vessels from the ostium to the distal vessels where the cross-sectional minimal area was < 2 mm2, regardless of the branching point, as previously described (JACC Cardiovasc Imaging. 2016 Sep;9(9):1074–83.). We have modified the sentence for more clarity. 

Line 229-231; The mean luminal attenuation (HU) was measured at 1 mm intervals, from the ostium to distal levels where the cross-sectional minimal area fell below 2 mm2 (12).

5. Line 185. I believe "distal levels" should be "distal vessels".

Answer: Thank you for this thoughtful comment. We have revised our manuscript accordingly.

Line 229-231; The mean luminal attenuation (HU) was measured at 1 mm intervals, from the ostium to distal levels where the cross-sectional minimal area fell below 2 mm2 (12).

6. Line 188. I believe "ostium length" should be changed to "vessel length".

Answer: Thank you for this thoughtful comment. We have revised our manuscript accordingly.

Line 232-234; TAG was defined as the linear regression coefficient between the intraluminal HU and the distance from the ostium. 

7. Line 266. Is there a minus sign missing from the baseline LAD TAG value?

Answer: Thank you for this thoughtful comment. We have revised our manuscript accordingly.

Line 345-348; The mean TAG values were significantly lower (steeper slop) for baseline CT than that for IV nitrate CT, especially for LAD and LCX (LAD, -20.92±8.37 vs -16.71±6.90; P<0.001; LCX, -29.56±17.37 vs -22.75±11.97; P=0.001; RCA, -11.50±7.08 vs -9.94±5.67; P=0.068; Table 3).

8. Was there a difference in coronary dominance between spasm positive and spasm negative patients?

Answer: Thank you for this comment regarding coronary dominance, which is a very important point in coronary researches. We only included right-side dominant patients since TAG can be influenced by the vessel diameter and length (Br J Radiol. 2018 Jul; 91(1087): 20180043). We have clearly explained this in the revised manuscript. 

Line 115-117; Since TAG could be influenced by the vessel diameter and length, we included only right-dominant patients i.e., patients whose posterior descending artery is supplied by the right coronary artery (RCA) (17). 

9. Figure 5. I believe these results are also reported in table 4. I would consider omitting table 4, and only use the figure. If you do this, you need to label the results of statistical analysis in the figure.

Answer: We appreciate your important comment regarding overlapping data. We decided to omit table 4, keep figure 5 (in revised manuscript, figure 6), and add additional statistical analysis in the figure (Line 382).

10. Figure 5. Panel B is incorrectly labeled as IV nitrate CT.

Answer: Thank you for this thoughtful comment. We have revised our manuscript accordingly.

.

11. Figure 5. Panel C is incorrectly labeled as baseline CT.

Answer: Thank you for this thoughtful comment. We have revised our manuscript accordingly.

12. Line 364, discussion. Please change "epithelial" to "endothelial".

Answer: Thank you for this thoughtful comment. We have revised our manuscript accordingly.

Line 476-480; Akasaka et al. (32) reported that vessels with diffuse spasms had significantly lower coronary flow reserves than those with focal spasms, which suggested that focal spasms are associated with localized endothelial dysfunction of the epicardial coronaries with no significant effect on coronary microvascular function. 

Reviewer #3: 

1. Subjects for this analysis were enrolled in the NAVIGATOR study but according to the references for that study, only 41 subjects with vasospastic angina were enrolled. The authors need to clarify the discrepancy with the reported 57 enrolled subjects. During what dates were subjects enrolled in the NAVIGATOR study?

Answer: Thank you for this thoughtful comment. Among the 57 enrolled participants, we excluded 8 patients with significant fixed stenosis and 6 patients who showed contradictory results on the CCTA (negative) and ergonovine provocation test (positive), which implies false negative results. Finally, we included 43 patients. The enrolled patients participated in the NAVIGATOR study from March 2017 to April 2019 rather than the study you mentioned (Eur Radiol. 2017 Mar;27(3):1136–47; Cardiology. 2018;139(1):25–32).

2. The authors excluded 6 subjects with invasive evidence of vasospasm in whom the CCTA did not reveal spasm (false negative). This is a major source of bias, and these subjects should be included in the analysis.

Answer: Thank you for this thoughtful comment. Accordingly, we have enhanced our discussion regarding false negative. The six patients showed negative results on CCTA but positive results on the ergonovine provocation test, which could be attributed to individual differences and variations in the timing of CT acquisition. Given the variations in the timing and migratory nature of vasopasms, CCTA studies on coronary spasms have generally discarded such false negative results; accordingly, we exclude them from the results.

Line 517-521; Third, we excluded patients with inconsistent results on CCTA (negative) and the ergonovine provocation test (positive). False-negative results could result from variations in the timing and migratory nature of vasospasms; moreover, they are generally excluded in related studies. 

3. Table 2 is incredibly confusing. I am assuming column 3 represents all vessels without and column 4 all vessels with spasm on CCTA. What one cannot tell from that table is how many vessels without spasm on invasive imaging had spasm on CCTA. If the goal of the table was to depict the proportion of spasm type (focal vs diffuse) in the specific arteries of patients with spasm detected by CCTA, a figure would likely be better. On a related note, what was the correlation spasm detected in the same arteries on invasive provocative testing and were the proportions of spasm type similar?

Answer: Thank you for this thoughtful comment. No vessels showed spasms on CCTA without spasms on the ergonovine provocation test. We modified table 2 more simply, and have added a diagram showing the proportion of spasm types in the specific arteries (Table 2 and Figure 4). We defined focal and diffuse type on CCTA; moreover, we classified spasm-positive vessels according to the aforementioned types.

4. Table 3 shows mean TAG and ProxHU results and per vessel differences before and after iv NTG. Subjects with and without spasm on invasive testing are lumped together but would it not be more useful to determine if the change in each parameter with NTG is different between subjects with and without invasive evidence of spasm?

Answer: Thank you for this thoughtful suggestion regarding the evaluation of TAG and ProxHU values according to the presence of spasms. Accordingly, we have added further results and statistical analyses. Regarding the comparison of the TAG values within the same spasm group [spasm (-) and (+) separately] between the baseline and IV nitrate CT, the spasm (+) group showed a larger difference in the mean TAG value than the spasm (-) group in the LCX and RCA [LCX, 43.5% vs. 19.4%; RCA, 22.8% vs. 9.7%; spasm (+) vs. spasm (-)]. Contrastingly, the LAD showed a smaller change in the TAG value in the spasm (+) group than in the spasm (-) group (14.4% vs. 24.8%). Compared with the LAD, the LCX and RCA showed a higher proportion of the diffuse type than the focal type (diffuse type: 100%, LCX; 80%, RCA; 67%, LAD), which could have contributed to our findings

Line 458-465; Comparison of TAG values in baseline CT between spasm(-) and (+) vessels revealed significant differences in LCX, but not in LAD and RCA. This could be attributed to the relatively hypoplastic nature of LCX, and the small sample size which only included diffuse spasm (+) vessels in the LCX. The spasm (+) group showed a larger mean TAG difference than the spasm (-) group for all coronary arteries except for LAD, which showed a smaller change in the spasm (+) group than in the spasm (-) group. Compared with LAD, LCX and RCA showed a higher proportion of the diffuse type than the focal type (diffuse type: 100%, LCX; 80%, RCA; 67%, LAD), which could have contributed to our findings. 

5. The information presented in Table 4 and Figure 4 are identical and therefore redundant. Furthermore, some of the same data is also presented in the text.

Answer: We appreciate your important comment regarding overlapping data. We decided to omit table 4, keep figure 5 (in revised manuscript, figure 6), and add additional statistical analyses in the figure.

6. The manuscript would be greatly improved if the authors correlated TAG (or more specifically the change in TAG with iv NRG) to the gold standard invasive provocative testing. Calculation of sensitivity/specificity and positive/negative predictive value would help determine if measuring TAG (or ProxHU) by CCTA is a useful non-invasive method to diagnose vasospastic angina.

Answer: Thank you for this important comment. Indeed, we desired to calculate more precise and meaningful results, including sensitivity or specificity. However, our primary goal was determining the relationship between the TAG value and spasm types. There were obstacles impeding more elaborate analysis, including the lack of data volume. Moreover, the TAG value only showed significant differences in the LCX, which further impeded analysis of sensitivity or specificity. Therefore, we focused on presenting differences in TAG values between spasm types, rather than calculating statistically precise values. However, we anticipate further research on the feasibility of TAG compared with conventional CAG in VA diagnosis. We have added an additional comment regarding this limitation.

Line 521-524; Fourth, TAG values showed significant differences only in the LCX. Therefore, we focused on presenting differences in the TAG values between the spasm types, rather than presenting the sensitivity or specificity of the CCTA or TAG values in VA diagnosis. 

Minor comments:

1. The authors need to better clarify what type vasospasm to which they are referring. The reader finds it hard to know if they are talking about spasm detected by invasive provocative testing of by CCTA.

Answer: Thank you for your comment. We defined the focal- and diffuse-type in CCTA; moreover, we classified spasm¬-positive vessels according to the aforementioned types.

Line 185-190; Regarding VA diagnosis, the following were the requirements for a positive finding on CCTA: (a) significant focal stenosis with negative remodeling on baseline CT without definite evidence of plaques in a completely dilated artery on IV nitrate CT (“focal-type”), or (b) diffuse small diameter (<2 mm) of a major coronary artery with a beaded appearance on baseline CT that showed complete dilation on IV nitrate CT (“diffuse-type”) (Figs 2 and 3) (10).

2. There are several grammatical errors that should be remedied.

Answer: The manuscript has been carefully rechecked; also we conducted an English editing service to improve the manuscript.

6. PLOS authors have the option to publish the peer review history of their article (what does this mean?). If published, this will include your full peer review and any attached files.

Answer: We agree to publish the peer review history of the article

---

## [Decision Letter · Decision Letter 1]

9 May 2022

PONE-D-21-39960R1Assessment of coronary spasms with transluminal attenuation gradient in coronary computed tomography angiographyPLOS ONE

Dear Dr. Kang,

Thank you for submitting your manuscript to PLOS ONE. After careful consideration, we feel that it has merit but does not fully meet PLOS ONE’s publication criteria as it currently stands. Therefore, we invite you to submit a revised version of the manuscript that addresses the points raised during the review process.

We look forward to receiving your revised manuscript.

Kind regards,

Ivana Isgum

Academic Editor

PLOS ONE

Additional Editor Comments (if provided):

The manuscript has been reviewed by three experts. While all Reviewers recognize the potential value of the work, Reviewer 3 pointed out to important limitations that were not sufficiently addressed in the revision and that need to be addressed before the manuscript can be considered for publication. I would ask the authors to carefully respond all issues raised by R3 and R4.

Reviewers' comments:

Reviewer's Responses to Questions

**Comments to the Author**

1. If the authors have adequately addressed your comments raised in a previous round of review and you feel that this manuscript is now acceptable for publication, you may indicate that here to bypass the “Comments to the Author” section, enter your conflict of interest statement in the “Confidential to Editor” section, and submit your "Accept" recommendation.

Reviewer #1: All comments have been addressed

Reviewer #3: (No Response)

Reviewer #4: All comments have been addressed

2. Is the manuscript technically sound, and do the data support the conclusions?

Reviewer #1: Yes

Reviewer #3: No

Reviewer #4: Yes

3. Has the statistical analysis been performed appropriately and rigorously? 

Reviewer #1: Yes

Reviewer #3: I Don't Know

Reviewer #4: Yes

4. Have the authors made all data underlying the findings in their manuscript fully available?

Reviewer #1: Yes

Reviewer #3: Yes

Reviewer #4: No

5. Is the manuscript presented in an intelligible fashion and written in standard English?

Reviewer #1: Yes

Reviewer #3: (No Response)

Reviewer #4: Yes

6. Review Comments to the Author

Reviewer #1: (No Response)

Reviewer #3: As mentioned in my previous review, the non-invasive diagnosis of patients with vasospastic angina is a worthy goal. As the authors point out in the Introduction “there have been no studies on TAG in coronary spasm” and therefore a study that adequately explores it utility would be useful. The hypothesis underlying this study is that “TAG for CCTA may allow a higher diagnostic performance for coronary spasm than CCTA alone.” Unfortunately, persistent major flaws in this study preclude confirming or refuting that hypothesis. These include:

1. The authors excluded 6 subjects with invasive evidence of vasospasm in whom the CCTA did not reveal spasm (false negative). This is a major source of bias, and these subjects continue to be excluded by the authors in this revised manuscript. One cannot accurately assess the diagnostic performance of using CCTA with TAG compared to invasive coronary angiography by excluding these subjects.

2. In the places in text and their rebuttal, the Authors appear to shift the purpose of the manuscript and narrow it to a description of the imaging features of coronary spam on CCTA. Given that the performance of CCTA with TAG in diagnosing coronary vasospasm has not been validated against invasive coronary angiography, an analysis of the type of spasm (focal versus diffuse) in this highly selected population is of limited value in an of itself. Furthermore, if that really is the main purpose of this work, the authors need to compare the spasm on CCTA to that observed on coronary angiography, the goal standard.

3. The revised Table 3 snow shows the difference in group mean TAG and ProxHU results. A far more useful metric is the per subject differences that could potentially be able to identify subjects with vasospasm.

Reviewer #4: The authors present an interesting study and have addressed the previously raised criticisms in a satisfactory manner. Two points remain:

1) I don't see an explicit statement that the data is made available. Please add this.

2) Please have the manuscript reviewed by a native speaker. There are still language issues.

7. PLOS authors have the option to publish the peer review history of their article (what does this mean?). If published, this will include your full peer review and any attached files.

Reviewer #1: No

Reviewer #3: No

Reviewer #4: No

---

## [Author Response · Author response to Decision Letter 1]

30 May 2022

Response to reviewers

Reviewer #1: 

Reviewer #3: 

As mentioned in my previous review, the non-invasive diagnosis of patients with vasospastic angina is a worthy goal. As the authors point out in the Introduction “there have been no studies on TAG in coronary spasm” and therefore a study that adequately explores it utility would be useful. The hypothesis underlying this study is that “TAG for CCTA may allow a higher diagnostic performance for coronary spasm than CCTA alone.” Unfortunately, persistent major flaws in this study preclude confirming or refuting that hypothesis. These include:

1. The authors excluded 6 subjects with invasive evidence of vasospasm in whom the CCTA did not reveal spasm (false negative). This is a major source of bias, and these subjects continue to be excluded by the authors in this revised manuscript. One cannot accurately assess the diagnostic performance of using CCTA with TAG compared to invasive coronary angiography by excluding these subjects.

Answer: 

Thank you for this valuable comment. In agreement, we have now included the six patients who showed negative results on CCTA but positive results on the ergonovine provocation test in the evaluation of the diagnostic performance of CCTA compared to invasive coronary angiography. The results of the per-patient analysis showed that the sensitivity, specificity, positive predictive value, negative predictive value, and accuracy were 80%, 100%, 100%, 76%, and 87.76%, respectively. We also analyzed that with per vessel level, and added the results with supplement table (Table S1). However, based on our results, there was only partial differences of TAG values for regarding the spasm on diffuse type spasm on LCX. Also we found the differences of TAG values between the spasm subtypes (diffuse vs focal). Thus, we could not applied TAG for the additional diagnostic tool of spasm detection on CCTA. We added this on the discussion session for limitation. 

Line 107-110; The diagnostic performance of CCTA for the detection of coronary spasm showed that the sensitivity, specificity, positive predictive value, negative predictive value, and accuracy were 80%, 100%, 100%, 76%, and 87.76%, respectively. Per-vessel analysis results are shown in S1 Table.

Line 448-452; Fourth, the results regarding the TAG differences between spasm(+) and (-) vessels, regardless of vessel and spasm types, could not support an improved VA diagnostic performance with the addition of TAG in the CCTA analysis. We therefore focused on presenting differences in the TAG values between the spasm types. 

2. In the places in text and their rebuttal, the Authors appear to shift the purpose of the manuscript and narrow it to a description of the imaging features of coronary spam on CCTA. Given that the performance of CCTA with TAG in diagnosing coronary vasospasm has not been validated against invasive coronary angiography, an analysis of the type of spasm (focal versus diffuse) in this highly selected population is of limited value in an of itself. Furthermore, if that really is the main purpose of this work, the authors need to compare the spasm on CCTA to that observed on coronary angiography, the goal standard. 

Answer: 

Thank you for your comment. As your comment, in the previous manuscript, we shifted the purpose from proving the hypothesis “that using TAG for CCTA may allow a higher diagnostic performance for coronary spasm than CCTA alone”, which was suggested in the introduction, to a description of the imaging features of coronary spam on CCTA. We agree with your comment that our results are insufficient to support the former purpose; therefore, we have deleted the sentence from the manuscript.

 In addition to CCTA alone, we attempted to evaluate the usefulness of TAG and TAG difference values in diagnosing VA in CCTA. Unfortunately, both the TAG and TAG differences between any of the groups (i.e. normal vs spasm(+) or diffuse type vs focal type) were not significant, except for the analysis of baseline CT TAG values among normal vs focal vs diffuse type in LAD and RCA (Table 4). Thus, in the revised manuscript we have focused on presenting the differences in TAG values between spasm types.

Line 430-436; Moreover, according to the present study, the differences in TAG values of coronary spasms depend on the morphological subtype, which may improve the detection of diffuse type spasm. However, due to numerous limitations, including the difficulty of performing CCTA twice for the same patient, variations in vasospasm timing, increased radiation exposure, increased amount of contrast material used, and the requirement of CAG for confirmatory diagnosis of VA, further research is needed to determine the clinical use of the double acquisition CCTA protocol.

Line 448-452; Fourth, the results regarding the TAG differences between spasm(+) and (-) vessels, regardless of vessel and spasm types, could not support an improved VA diagnostic performance with the addition of TAG in the CCTA analysis. We therefore focused on presenting differences in the TAG values between the spasm types. 

3. The revised Table 3 snow shows the difference in group mean TAG and ProxHU results. A far more useful metric is the per subject differences that could potentially be able to identify subjects with vasospasm. 

Answer: 

Thank you for your comment. Accordingly, we calculated the TAG difference values, the intra-subject difference in TAG value between the baseline CT and IV nitrate CT. The results showed non-significant differences in TAG difference values among normal, focal spasm, and diffuse spasm groups in all three vessels. Among vessel subtypes, LCX, of which all spasm(+) vessels were diffuse type, showed a greater TAG difference for diffuse spasm(+) than for normal type, although this was not statistically significant.

Line 349-357; The intra-subject difference in TAG value between the baseline CT and IV nitrate CT shows relatively larger value in diffuse type spasm vessels than in focal type spasm or negative spasm vessels, but statistically insignificant (diffuse type vs focal type vs normal, 6.20±13.77 vs 2.05±6.13 vs 3.83±7.04; P=0.35; Table 5). Among vessel subtypes, LCX, of which all spasm(+) vessels were diffuse type, showed a greater TAG difference for diffuse spasm(+) than for normal type, although this was not significant (diffuse type vs normal, 15.11±25.68 vs 5.38±7.07; P=0.052; Table 5). The other two vessels, LAD and RCA, did not show significant TAG difference values among normal, focal spasm, and diffuse spasm vessels (LAD, P=0.326; RCA, P=0.433; Table 5).

Line 397-399; Moreover, the intrasubject TAG difference in LCX was greater in the diffuse spasm group than in the normal group. This could be attributed to the relatively hypoplastic nature of LCX, and the small LCX sample size which only included diffuse spasm(+) vessels.

Reviewer #4: 

The authors present an interesting study and have addressed the previously raised criticisms in a satisfactory manner. Two points remain:

1. I don't see an explicit statement that the data is made available. Please add this. 

Answer: Thank you for your comment. All relevant data are within the manuscript, figures, and its Supporting Information files.

2. Please have the manuscript reviewed by a native speaker. There are still language issues.

Answer: Thank you for your comment. The manuscript has now been proofread and corrected by a professional editor.

---

## [Decision Letter · Decision Letter 2]

27 Jun 2022

Assessment of coronary spasms with transluminal attenuation gradient in coronary computed tomography angiography

PONE-D-21-39960R2

Dear Dr. Kang,

We’re pleased to inform you that your manuscript has been judged scientifically suitable for publication and will be formally accepted for publication once it meets all outstanding technical requirements.

Kind regards,

Matteo Tebaldi

Academic Editor

PLOS ONE

Additional Editor Comments (optional):

Reviewers' comments:

Reviewer's Responses to Questions

**Comments to the Author**

1. If the authors have adequately addressed your comments raised in a previous round of review and you feel that this manuscript is now acceptable for publication, you may indicate that here to bypass the “Comments to the Author” section, enter your conflict of interest statement in the “Confidential to Editor” section, and submit your "Accept" recommendation.

Reviewer #1: All comments have been addressed

Reviewer #4: All comments have been addressed

2. Is the manuscript technically sound, and do the data support the conclusions?

Reviewer #1: Yes

Reviewer #4: (No Response)

3. Has the statistical analysis been performed appropriately and rigorously? 

Reviewer #1: Yes

Reviewer #4: (No Response)

4. Have the authors made all data underlying the findings in their manuscript fully available?

Reviewer #1: Yes

Reviewer #4: (No Response)

5. Is the manuscript presented in an intelligible fashion and written in standard English?

Reviewer #1: Yes

Reviewer #4: (No Response)

6. Review Comments to the Author

Reviewer #1: The authors have addressed my previous comments and concerns in a satisfactory manner.

Reviewer #4: (No Response)

7. PLOS authors have the option to publish the peer review history of their article (what does this mean?). If published, this will include your full peer review and any attached files.

Reviewer #1: No

Reviewer #4: No

---

## [Editor Report · Acceptance letter]

1 Jul 2022

PONE-D-21-39960R2 

Assessment of coronary spasms with transluminal attenuation gradient in coronary computed tomography angiography 

Dear Dr. Kang:

I'm pleased to inform you that your manuscript has been deemed suitable for publication in PLOS ONE. Congratulations! Your manuscript is now with our production department. 

Kind regards, 

on behalf of

Dr. Matteo Tebaldi 

Academic Editor

PLOS ONE